# Mucosal Sporotrichosis from Zoonotic Transmission: Descriptions of Four Case Reports

**Yong Yaw Yeow [1],\*, Xue Ting Tan [2] and Lee Lee Low [1]**

[1] Infectious Diseases Unit, Department of Medicine, Hospital Sultanah Bahiyah, Alor Setar 05460, Malaysia
[2] Infectious Diseases Research Center, Institute for Medical Research, National Institutes of Health, Kuala Lumpur 50588, Malaysia
\* Correspondence: yongyaw1210@gmail.com

**Abstract:** Background: Sporotrichosis is a subacute or chronic mycosis caused by a dimorphic fungus of the genus *Sporothrix*. Zoonotic-transmitted sporotrichosis has become a major public health concern and is characterised by a different clinical pattern from the traditional epidemiology of sporotrichosis. Case presentation: We present the details of four patients with mucosal sporotrichosis with regional lymphadenopathy (three cases of granulomatous conjunctivitis and one case of nasal sporotrichosis). The patients' age range was between 23 to 46 years old and their gender was three female and one male patient. All four patients shared the same ethnicity, Malay, and they had a common history of owning domestic cats as pets. *Sporothrix schenckii* were isolated from all the culture samples and its antifungal susceptibility patterns were compared in the mycelial and yeast phases. All four patients recovered with oral itraconazole treatment, but the treatment duration was variable among patients. Conclusions: People who have a history of contact with domestic cats should be aware of the possibility of sporotrichosis infection. It can present in cutaneous, lymphocutaneous, disseminated, or systemic forms. Early treatment and the prevention of disease progression are more beneficial to patients. The published data concludes that antifungal treatment is highly efficacious, although the reported treatment duration is variable.

**Keywords:** sporotrichosis; granulomatous conjunctivitis; parinaund oculoglandular syndrome; case report

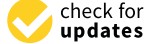



## 1. Introduction

Sporotrichosis is an often neglected subcutaneous or lymphocutaneous fungal infection caused by *Sporothrix schenckii*. A dimorphic organism, it was traditionally thought to be an occupational disease mostly seen in gardeners or horticulturists. Since the disease is primarily confined to tropical areas and it is not usually fatal, sporotrichosis has never been regarded as a major fungal threat to humans, unlike candidiasis, aspergillosis, and cryptococcosis. Zoonotic sporotrichosis caused by *Sporothrix* spp. has been documented widely in the United States, Malaysia, India, and Mexico [1]. In recent decades, there has been a change in the spectrum of disease, its epidemiology, and the disease burden. Sapronotic transmission from the environment (associated with traumatic inoculation through soil, plants, and decaying wood) is the most common source of human sporotrichosis, *S. schenckii*, but zoonotic infections have become increasingly common with the emergence of *S. brasiliensis*. *S. schenckii complex* consists of five distinct species—namely, *S. schenckii*, *S. globosa*, *S. mexicana*, *S. luiriei*, and *S. brasiliensis*. *S. schenckii*, *S. globosa*, and *S. brasiliensis* are able to infect both humans and animals [2]. Sporotrichosis has emerged as a zoonotic disease with a potential threat to public health, with increasing reports of domestic animals (mainly dogs and cats) affected by sporotrichosis and implicated as dispersers of the fungi in the environment and domestic space.

## 2. Case Presentations

### 2.1. Case 1

A 28-year-old Malay lady with underlying allergic rhinitis and ectopic dermatitis presented with a 2-month history of progressively worsening unilateral swelling of the left eye. The swelling was associated with intermittent fever and an ipsilateral 2 cm × 2 cm lymph node swelling over the supraclavicular region. She has a cat as her companion animal at home. During admission, ophthalmic examination revealed multiple nodular granulomas over the left upper and lower tarsus, as well as bulbar conjunctiva (Figure 1). Her left eye visual acuity was 6/9. Examination of the right eye was normal with no observed cutaneous lesions, while systemic examination was also unremarkable. In line with the working diagnosis of Parinaud oculoglandular syndrome secondary to cat-scratch disease, oral azithromycin was initiated, but was later substituted with doxycycline due to allergic rashes post-azithromycin administration. The patient was also prescribed topical dexamethasone-ciprofloxacin eye drops. After an initial brief transient response, the lesions in her left eye recurred and progressed. HIV, Diabetes Mellitus, Autoimmune screenings, Bartonella, and scrub typhus serology were negative. Blood fungal Culture was sent and negative. Fine needle aspiration for cytology (FNAC) of the left supraclavicular lymph node was performed and the cytologic report revealed granulomatous lymphadenitis. The histopathology report of the left conjunctiva biopsy further corroborated the findings of chronic granulomatous inflammation. Conjunctival tissue for pan-fungal and mycobacterial tuberculosis PCR were both negative. The diagnosis was eventually secured when the fungus identified as *Sporothrix schenckii* complex, isolated from the conjunctival tissue culture by matrix-assisted laser desorption ionization–time of flight mass spectrometry (MALDI–TOF). The identification was confirmed with the molecular method by our reference laboratory, the Institute of Medical Research (IMR). Antifungal susceptibility patterns in both phases are illustrated in Table 1. Suspension itraconazole with a loading dose of 200 mg thrice per day for 3 days was prescribed, followed by 400 mg per day in addition to amphotericin B eye drops. Outpatient review at 28 weeks of treatment showed resolved lymphadenopathy, fully recovered left eye vision and residual tiny nodules at the inferior fornix of the left eye. Clinically, she had no complaints of any special discomfort. Repeat conjunctival biopsies for culture were sterile. Itraconazole was then discontinued after a total of 8-months treatment. Her cat was diagnosed with sporotrichosis and subjected to the same treatment regime following a veterinary consult (Figure 1).

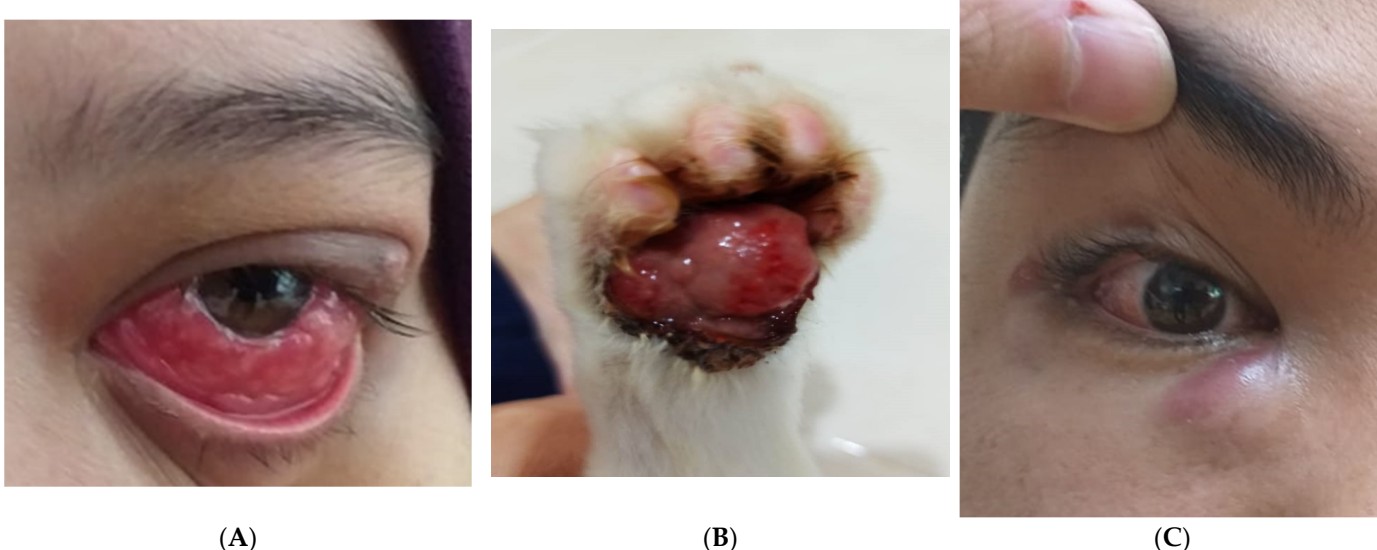

| (A) | (B) | (C) |

**Figure 1.** (**A**). Case 1-Granulomatous conjunctivitis of the left eye. (**B**). Infected paw—pet of Patient 1. (**C**). Case 3—Granulomatous conjunctivitis of the right eye with a small granuloma near the limbus.

**Table 1.** MIC in μg/mL for eight antifungal agents in both the mycelial and yeast phase.

| | | TRB | ITC | KTC | VRC | PSC | RVC | ISC | AMP |
|---|---|---|---|---|---|---|---|---|---|
| Case 1 | MP | 0.25 | <0.0313 | 0.25 | >16 | 0.125 | 0.5 | 0.25 | 4 |
| | YP | ≤0.0313 | ≤0.0313 | ≤0.0313 | ≤0.0313 | ≤0.0313 | ≤0.0313 | ≤0.0313 | 0.5 |
| Case 2 | MP | 0.0625 | 0.0313 | 0.0313 | 4 | ≤0.0313 | 0.0625 | 0.0313 | 4 |
| | YP | ≤0.0313 | ≤0.0313 | ≤0.0313 | ≤0.0313 | ≤0.0313 | ≤0.0313 | ≤0.0313 | ≤0.0313 |
| Case 3 | MP | 0.5 | 0.0313 | <0.0313 | >16 | ≤0.0313 | 0.0313 | 0.0625 | 8 |
| | YP | ≤0.0313 | ≤0.0313 | ≤0.0313 | ≤0.0313 | ≤0.0313 | ≤0.0313 | ≤0.0313 | ≤0.0313 |
| Case 4 | MP | 0.5 | 0.5 | 0.0625 | 16 | 0.5 | 1 | 2 | 1 |
| | YP | ≤0.0313 | ≤0.0313 | ≤0.0313 | ≤0.0313 | ≤0.0313 | ≤0.0313 | ≤0.0313 | ≤0.0313 |

TRB, Terbinafine; ITC, Itraconazole; KTC, ketoconazole; VRC, Voriconazole; PSC, Posaconazole; RVC, Ravuconazole; ISC, Isavuconazole; AMP, Amphotericin B, MP, mycelial phase, YP, yeast phase. *S. schenckii* moulds were inoculated on Potato Dextrose Agar (PDA) at 25 °C for 5–7 days. To obtain its yeast form, the *S. schenckii* mould was sub-cultured on Brain Heart Infusion (BHI) agar for several passages. The inoculated agar was incubated at 37 °C for 5–7 days. The inoculum size for the conidia mould and yeast were $0.4 \times 10^4$ to $5 \times 10^4$ CFU/mL and $5.0 \times 10^2$ to $2.5 \times 10^3$ CFU per mL, respectively.

### 2.2. Case 2

A 46-year-old premorbid-well, cat-loving Malay lady presented with 3 days' history of unilateral right blood-stained nasal discharge, fever, and headache. On examination, there was a uniform swelling of the right nose extending to the cheek and an isolated lymphadenopathy at the right cervical region. Rhinoscopic examination revealed multiple ulcers at the right anterior end of the inferior turbinate. There was no active bleeding or pus discharge. Computer tomography (CT) imaging of the paranasal sinuses showed ethmoidal and right maxillary sinusitis with inflammation of the right inferior turbinate and nasal ala, while sparing the bone and deep tissue structures. Histopathology of both the ethmoidal and maxillary sinus FNAC showed chronic granulomatous inflammation. The samples were absent of organisms with acid-fast bacilli (AFB) and periodic acid-schiff (PAS) staining. *Sporothrix schenkii* grew on sabouraund agar from the swab cultures. HIV and Diabetes mellitus screening were negative. Blood fungal cultures were sent and were negative. A diagnosis of nasal sporotrichosis was made. An oral itraconazole regime was initiated for the patient with a loading dose of 200 mg thrice per day for 3 days, followed by 400 mg per day. In the outpatient assessment 4 weeks post-treatment, her facial swelling and cervical lymphadenopathy had totally resolved. She experienced complete remission of all symptoms. Repeated rhinoscopic examinations showed complete resolution of earlier inflammation. She continued with oral itraconazole for a total duration of 8 weeks. On further questioning, she recalled an occasion when one of her cats sneezed onto her face while she was carrying it. Her two cats were subsequently diagnosed to have sporotrichosis and were subjected to oral itraconazole treatment.

### 2.3. Case 3

A 23-year-old Malay gentlemen with underlying diabetes mellitus, which was well controlled, presented with a 3-week history of a painless and inflamed right eye; this was associated with blurring of vision, itchiness, and a colourless discharge. On examination, he had a hyperemic right conjunctiva, with a visible granuloma overlying the lower and upper bulbar conjunctiva adjacent to the limbus. His visual acuity was markedly reduced over his right eye (6/12). A 2 cm lymph node was palpable over the right preauricular region. Examinations of other systems were unremarkable. HIV screening was negative. Blood fungal cultures were sent and were negative. The patient was prescribed with a course of oral azithromycin and topical betamethasone-neomycin. Despite this, he experienced a new erythematous nodule at the medial angle of the same eye measuring 1.5 cm × 1.5 cm. Ophthalmology services undertook a conjunctival biopsy while escalating treatment to a combination of oral doxycycline, topical amphotericin B, and ciprofloxacin eye drops.

Histopathological examination of the biopsied tissue showed suppurative granulomatous inflammation with no fungal bodies. Microbiological culture of the tissue grew *Sporothrix schenckii*. Intravenous amphotericin B was initiated, but ceased when computed tomography images of the brain and orbit excluded invasive disease. Suspension itraconazole was started with a loading dose of 200 mg thrice per day for 3 days, followed by a maintenance dose of 400 mg/day for two weeks, before switching to capsule itraconazole at 400 mg/day. An outpatient clinic review 5 months later showed resolution of both the granulomas and lymph node swelling. Itraconazole was stopped after 6 months of treatment and the patient has remained in remission. His diabetes mellitus was well controlled throughout the follow-up; Hb A1c ranged between 5.8–6%. The patient owns five cats, but one of them developed skin lesions and died 2 months prior to his owner's presentation.

*2.4. Case 4*

A 23-year-old Malay lady presented to the outpatient clinic complaining of a left eye swelling and a swollen neck, which appeared 1 week ago. She applied antibiotic eye drops purchased over the counter to no avail. Slit-lamp examination by ophthalmology services showed a nodular granuloma at the left lower and upper tarsus, with similar nodular lesions seen in her right eye at the upper and lower tarsus. The visual acuity of both eyes was normal. There were multiple enlarged lymph nodes over the left preauricular, post auricular, and supraclavicular regions. Her HIV and Diabetes Mellitus screening were negative. Blood fungal cultures were sent and were negative. Systemic and topical antibiotics were initiated. An excision biopsy of the left eye conjunctiva was performed. The histopathological report showed granulomatous conjunctivitis, while cultures grew *Sporothrix schenckii*. A course of oral itraconazole was initiated with a loading dose of 200 mg thrice per day for 3 days, followed by 400 mg per day concurrently with topical amphotericin B. An outpatient reassessment one month into treatment showed marked improvement of the eye swelling, while the lymphadenopathy had completely resolved. The patient experienced complete resolution of all symptoms 5 months into treatment and completed a total of 6 months of oral itraconazole. One of her cats died two months prior to her presentation while the other four remain healthy. She was advised to seek veterinary advice for her cats.

**3. Discussion and Conclusions**

*Sporothrix schenkii* is a dimorphic fungus commonly found in soil, plants, hay, and decaying wood. Historically, sporotrichosis has been associated with gardeners and agricultural workers—hence the term 'Rose gardener's disease'. Studies conducted in Brazil confirmed zoonotic transmission from domestic and wild animals via penetration of the skin or mucosa and inhalation [3]. Clinical presentations are sub-classified into fixed cutaneous, lymphocutaneous, disseminated, or systemic forms including pulmonary or osteoarticular sporotrichosis, meningitis, and endophthalmitis—both exogenous and endogenous. Lymphocutaneous manifestation is the most common presentation, often presenting either as a primary ulcerative lesion or as a subcutaneous nodule extending along the lymphatic drainage in a linear pattern.

All four patients shared a common history of contact with infected domestic cats. The initial working diagnosis was cat-scratch disease, but there was no relief with antibiotic therapy. To conclude the diagnosis, a tissue biopsy was obtained for histopathological examination and culture. Consistent with our patients described here, histopathological examination of the cutaneous or mucosal nodules only demonstrates non-specific nodular vasculitis [4] or granulomatous inflammation with no fungal bodies, leading to inconclusive findings. As such, it is best to obtain tissues for diagnostic cultures, antifungal susceptibility testing, and molecular sequencing to identify the species. The samples were outsourced to the Central National laboratory, Institute for Medical Research (IMR), for molecular methods to confirm the identification of *Sporothrix schenckii*, as the molecular method is unavailable in our local hospital laboratory.

The minimal inhibitory concentration (MIC) values obtained for all four isolates in both phases are presented in Table 1. The MICs of *S. schenckii* were obtained according to the broth microdilution method in Clinical and Laboratory Standards Institute (CLSI) M38 for the mould and CLSI M27 for the yeast form. The MICs of itraconazole, voriconazole, posaconazole, isavuconazole, and amphotericin B were read as the lowest drug concentration required to inhibit 100% of visible growth. The MIC of terbinafine and ketoconazole were read as the lowest concentration to inhibit 50% of visible growth. As a thermos-dimorphic fungi, it produces different anti-fungal susceptibilities between the mycelial (MP) and yeast phase (YP). Consistent with previous studies by Trilles et al. [5], significantly higher MIC values were observed in the mycelial than in the yeast phase. Interestingly, apart from amphotericin B, all antifungal MIC values were exceptionally low when tested in the YP. Higher expression of melanin in the filamentous form, differing in temperature and inoculation sizes, may play a role in the varying MIC values. In our study, the lowest MICs were found with itraconazole and ketoconazole, followed by terbinafine and posaconazole in the MP. Although terbinafine has high activity against *Sporothrix schenckii*, its treatment potential is limited to lymphocutaneous disease, and the optimal treatment dose is not well defined.

Our first three cases demonstrated low MIC values for ravuconazole and isavuconazole; this is in contrast with previous studies that have shown ravuconazole and isavuconazole MICs against sporothrix to be considerably higher [6,7]. Based on the MIC of amphotericin B in the MP, the isolate from case 3 might belong to a non-wild type strain with an MIC above the epidemiological cut-off values of 4 µg/mL [8]. This phenomenon has been reported by Han et al. [9], where isolates from infected felines demonstrated amphotericin B MIC values > 4 µg/mL (geometric mean MIC of 9.8)—an intriguing finding which warrants further exploration.

Antifungal susceptibility tests for the yeast form of dimorphic fungi have not been extensively validated, and the mycelium-to-yeast transformation is a laborious process. To date, there is a paucity of efficacy data on new azoles for the treatment of sporotrichosis. Further studies are needed to determine the efficacy of new antifungal agents for sporotrichosis treatment and the best method of elucidating the susceptibility profiles of dimorphic fungi in the yeast phase.

We looked through English articles from PubMed using the keywords 'granulomatous conjunctivitis' (GC) and 'nasal sporotrichosis' (NS). There were 83 case reports on granulomatous conjunctivitis and nine reports on nasal sporotrichosis. Of these, 28 cases of GC had concomitant lymphadenopathy and cutaneous nodules, 17 cases had involvement of lymph node(s), and six cases showed cutaneous lesions. Only one GC sporotrichosis had central nervous system involvement [10]. In contrast, ocular sporotrichosis such as endophthalmitis, uveitis, or retinal-choroiditis is frequently associated with osteomyelitis, intracranial abscess, endocarditis, and distant dissemination to other organs. Out of the 66 patients who were available for follow-up assessment, 100% of patients had a complete or partial response, while three patients developed symblepharon as a sequelae. Among studies where patients were reported as having nasal sporotrichosis, one had locally invasive disease with destruction of the uvula and hard palate, four had bony involvement, three had sinusitis, and another three had extended to the neurological system. Clinical manifestations of mucosal sporotrichosis and courses of treatment with outcomes reported in PubMed using the keywords 'granulomatous conjunctivitis' and 'nasal sporotrichosis' are summarised in Table 2 [3,10–38].

**Table 2.** Clinical manifestations of mucosal sporotrichosis and courses of treatment with outcomes reported in PubMed using the keywords 'granulomatous conjunctivitis' and 'nasal sporotrichosis'.

| Author | Number of Cases Reported | Presentation of Granulomatous Conjunctivitis | Enlarged Lymph Nodes | Cutaneous Involvement | Dissemination | Treatment | Duration | Outcomes |
|---|---|---|---|---|---|---|---|---|
| Xavier et al. [3] | 1 | Granulomatous conjunctivitis | No | Yes | No | Itraconazole 200 mg/day | 3–4 weeks after regression | Complete resolution |
| Arinelli et al. [10] | 26 | Granulomatous conjunctivitis | Yes 5 No 21 | Yes 5 No 21 | 1 CNS | 14 Itraconazole 100 mg/day 11 Itraconazole 200 mg/day 1 itraconazole 800 mg/day→ Posaconazole | 8 to 20 weeks Median 14 weeks | 23 complete resolutions 3 lost to follow up |
| Heidrich et al. [11] | 1 | Nasal nodular | No | No | No | Potassium iodine 20 drops TDS Potassium iodine 25 drops TDS Itraconazole 200 mg/day Terbinafine 500 mg/day | 3 months 11 months 5 months | Relapse Relapse Cured |
| Ribeiro et al. [12] | 10 | Granulomatous conjunctivitis | 7 Yes 3 No | No | No | Itraconazole 200 mg/day | 3–6 months | Favourable response |
| Ramírez Soto et al. [13] | 21 | 20 Granulomatous conjunctivitis 1 granulomatous conjunctivitis plus nodule-ulcerated on the face | 13 yes 8 No | 13 Yes 8 No | No | Potassium iodide 4 to 40 drops/3 times daily in adults | Median 45 days (30–60 days) | 21 10 cured 10 dropped out 1 not treated |
| McGrath et al. [14] | 1 | Granulomatous conjunctivitis | Yes | Yes | No | Aureomycin 250 mg od × 4 days Potassium iodine 5 to 30 drops tds | 4–5 months | Complete resolution |
| Lacerda Filho et al. [15] | 1 | Granulomatous conjunctivitis | Yes | No | No | Itraconazole 200 mg/day | 3 months | Responding |
| Paiva et al. [16] | 1 | Granulomatous conjunctivitis | Yes | Yes | No | Itraconazole 200 mg/day | 3 months | Responding |
| Alvarez et al. [17] | 1 | Granulomatous conjunctivitis | Yes | No | No | Potassium iodine 30 drops TDS | 90 days | Complete resolution |
| Gordon et al. [18] | 1 | Granulomatous conjunctivitis | No | Yes | No | Potassium iodine 25 drops qid | 7 weeks | Healed with scar |
| Medeiros et al. [19] | 1 | Granulomatous conjunctivitis | Yes | Yes | No | Itraconazole 200 mg/day | 60 days | Complete resolution |
| Lemes et al. [20] | 2 | Granulomatous conjunctivitis | Yes 1 No 1 | Yes 2 | No | Itraconazole Unspecified dose | Unspecified | Complete resolution |
| Ramírez-Oliveros et al. [21] | 3 | Granulomatous conjunctivitis | Yes No 3 | Yes 2 No 1 | No | Itraconazole 200 mg/day 100 mg/day | 3 months 5 months | Complete resolution |
| Liborio Neto et al. [22] | 1 | Granulomatous conjunctivitis | Yes | No | No | Itraconazole 100 mg/day | 3 months | Complete resolution |
| Castro et al. [23] | 1 | Ulcerated, extropia, Left eye blindness Nasal destruction | No | Yes | Yes—Osteolytic lesion in hand–wrist–forearm bone | Amphotericin b | Total dose 2450 mg Unspecified duration | Persistent osseous lesion Regression of cutaneous lesion |
| Ling et al. [24] | 1 | Granulomatous conjunctivitis | Yes | no | No | Itraconazole 400 mg/day | 6 months | Complete resolution |
| Sun et al. [25] | 1 | Granulomatous conjunctivitis Dacryocystitis | No | Yes | No | Itraconazole 400 mg/day | 3 months | Complete resolution |
| Biancardi et al. [26] | 3 | Multifocal choroiditis. Nasal mucosa | Yes 1 No 2 | Yes | Yes— Osteomyelitis Bacteraemia | Amphotericin b 1 mg/kg/day Itraconazole 200–400 mg/day | 1–30months | cured |
| Zhang et al. [27] | 3 | Nasal mucosa | No | Yes | No | Potassium iodine 10% 0.25–1 g/day +_ itraconazole 50 mg/day | 3–4 months | Complete resolution |
| MF Matos et al. [28] | 1 | Granulomatous conjunctivitis Lacrimal gland | Yes | Yes | No | Amphotericin b 50 mg/day Itraconazole 200 mg/day | Unspecified 8 months | symblepharon in the lower temporal region complete ptosis |
| Gameiro Filho et al. [29] | 1 | Granulomatous conjunctivitis Lacrimal apparatus | Yes | Yes | No | Itraconazole 200 mg/day | 3 months | Complete response |

**Table 2.** *Cont.*

| Author | Number of Cases Reported | Presentation of Granulomatous Conjunctivitis | Enlarged Lymph Nodes | Cutaneous Involvement | Dissemination | Treatment | Duration | Outcomes |
|---|---|---|---|---|---|---|---|---|
| Yamagata et al. [30] | 3 | 1. Granulomatous conjunctivitis 2. Granulomatous conjunctivitis 3. Granulomatous conjunctivitis | No Yes Yes | Yes Yes No | No No No | Itraconazole 200 mg OD Itraconazole 200 mg OD × 2/12 then 400 mg 6/12 Itraconazole 200 mg OD | 9 months 8 months 15 days | Complete response |
| Schubach et al. [31] | 2 | Granulomatous conjunctivitis | Yes | Yes | No | Itraconazole 100 mg OD | 3 months | Complete response |
| Hampton et al. [32] | 1 | Granulomatous conjunctivitis | Yes | No | No | Itraconazole 200 mg OD 2 week then 300 mg OD | 4 months | Complete resolution |
| Eyer-Silva et al. [33] | 1 | Nasal septum perforation Palate ulcer Uvular ulcer | No | No | No | Amphotericin b 35 mg/day Itraconazole 400 mg/day | 30 days >1 year unspecified | Resolution of the hard and soft palate ulcers, full destruction of the uvula, retraction of the right ala nasi |
| Freitas et al. [34] | 21 | 3 Nasal septum destruction 1 Granulomatous conjunctivitis | Yes 7 | Yes 21 | Yes 2 Bone involved 2 Meningitis | Amphotericin b 1–2.5 g Itraconazole 100–400 mg/day | 1–11 months 1–12 months | 16 cured 2 deaths 2 lost to follow up 1 relapse Remark: 1 eye cured with itraconazole 200 mg/day for 6 months 3 nasal cured with 2 amphotericin b 2–2.5 g 3–5 months 1 Itraconazole 100 mg/day for 11 months |
| Anita et al. [35] | 1 | Nasal septum perforation Maxillary sinus mass nodule | No | No | No | Itraconazole 400 mg/day | 3 months | Cured |
| Kumar et al. [36] | 1 | Nasal septal perforation Lateral wall and turbinates destruction Pansinusitis with polyposis | No | No | Yes— Intracranial, Intraorbital | Removal polyposis Itraconazole 200 mg/day Potassium iodine 60 drops tds | 3 months 6 months | Complete resolution |
| Morgan et al. [37] | 1 | Right inferior nasal turbinate destruction Maxillary sinusitis | No | No | No | Itraconazole 400 mg/day | 1 year | Passed away due to other cause No evidence that the infection had recurred |
| Ferreira et al. [38] | 1 | Nasal mucosa Oral mucosa | No | Yes | Yes Lungs | Liposomal amphotericin b 100 mg/day Itraconazole 200 mg/day | 3 months | Passed away |
| Noguchi et al. [39] | 1 | Nose (dorsum) | Yes | Yes | No | Itraconazole 100 mg/day | 16 weeks | Healed |

With regard to treatments for conjunctivitis and nasal sporotrichosis, systemic literature analysis showed that a saturated solution of potassium iodine (SSKI) and Itraconazole were the most commonly used agents. Amphotericin B was the agent of choice in disseminated disease. Treatment responses to Itraconazole or SSKI were overall satisfactory—only Heldrich et al. [11] reported a relapse of disease while on SSKI. Posaconazole was used on a patient with GC sporotrichosis complicated with neurological extension [10]. Spontaneous remissions were reported in two isolated cases of uncomplicated GS and NS, respectively [40,41]. The treatment duration varied between individuals, ranging from 4 weeks to 6 months in GS, while the majority of patients received 3 months of antifungal

treatment. As for NS, treatment duration was from 4 weeks to 4 months, and could extend to 12 months in case of distant dissemination.

Our case series illustrates uncommon presentations of sporotrichosis with conjunctivitis and nasal ulcers. Given a recent history of contact with feline animals, sporotrichosis has to be considered as one of the differential diagnoses with a high index of suspicion. This further confirms the animal-to-human transmission of sporotrichosis. Both GS and NS rarely resolve spontaneously—as such, systemic antifungal agents should be administered accordingly to prevent worsening or relapse of disease. As described in Case 1, patients may initially respond to topical anti-inflammatory agents, but could soon experience worsening of earlier symptoms. In a patient with intranasal sporotrichosis, ascertaining an invasive fungal infection of the sinuses is necessary to determine the disease extent. Itraconazole was chosen as the antifungal in our cases. Nonetheless, Therapeutic drug monitoring (TDM) of itraconazole was not performed in our four patients in view of it not being available in our centre. Unfortunately, there were no samples taken from the feline pets in our patients because of there being no linkage of care for animals between the veterinarians and clinicians in our setting. This results in limitations in diagnosing feline infectious diseases. However, the *Sporothrix shenckii* complex has been isolated from cats in Malaysia and sequencing analysis has proven that clinical isolates from humans and animals are identical [42,43].

*Sporothric schenkii* is intrinsically resistant to many antifungal agents. It is important to identify the *Sporothrix* isolate to the species level, as the antifungal susceptibility profiles vary between species and geographical areas. The different grades of virulence among species are influenced by the presence of melanine, glycoproteins, and other cell-wall components. As such, species identification plays an important role in ascertaining outcomes and treatment choices.

Although the optimal treatment duration has not been established, generally, a prolonged 3 to 6-month treatment course is recommended, and therapy should only be terminated two to four weeks after the complete resolution of ocular or nasal lesions [44]. Sporotrichosis is a zoonotic disease with public health importance and is an occupational hazard for animal handlers. Until a *Sporothrix* vaccine is available, cat owners and veterinarians must take precautions to prevent bites or scratches from animals. Owners should take measures to avoid stray animals from mixing with their pets, and should bring their pets for a veterinary consult when there is persisting, non-healing wound. Greater collaboration in managing infected pets is necessary to ensure timely diagnosis and treatment of both pets and humans.

**Author Contributions:** Y.Y.Y. and L.L.L. collected the data from the patients, followed the patients, and prospectively recorded the patients' clinical data. X.T.T. isolated the fungus and performed molecular sequencing and antifungal susceptibility testing. All authors have read and agreed to the published version of the manuscript.

**Funding:** There was no funding or other financial support from any organizations.

**Institutional Review Board Statement:** This study was conducted according to the guidelines of Declaration of Helsinki, and was approved by Medical Research and Ethics Committee and National Medical Research Register Malaysia. The patients signed an informal consent form.

**Informed Consent Statement:** The data and materials, including the clinical data of the patients, are included within the article. The datasets are available from the first author.

**Data Availability Statement:** Written informed consent was obtained from the patients for the publication of these case reports and any accompanying images. A copy of the written consent is available for review by the Editor-in-Chief of this journal.

**Acknowledgments:** We would like to acknowledge the doctors and scientific officers of the Institute Medical Research, Malaysia who conducted the molecular and antifungal susceptibility tests—not forgetting the physicians, nurses, and technicians who are not listed as authors, but have contributed to the care of our patients.

**Conflicts of Interest:** The authors declare no conflict of interest.

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
