# Peer review of "Mucosal Sporotrichosis from Zoonotic Transmission: Descriptions of Four Case Reports"

_2036-7449, doi:10.3390/idr15010011_

Round 1

Reviewer 1 Report

The study authors report on 4 cases of zoonotic sporotrichosis with common denominators of owning cats as pets and being of Malay ethnicity. They describe the clinical histories of their patients and elaborate on the antifungal susceptibility testing results and their clinical implications. They also follow it with a thorough review of the reported cases with mucosal and conjunctival involvement.

Overall, I believe the manuscript is well written and describes a unique cohort of patients, with an exhaustive discussion. It will help raise awareness about this unique disease manifestation and epidemiologic link in clinicians. I have a few points to add as below.

Major Comments

1. Case 1, 2 and 4 – was there any testing performed to evaluate for underlying diabetes mellitus (DM) or HIV? Additionally, given the age of patient #3, was HIV testing discussed and performed? It would be helpful to include this information. Additionally, in all the patients, it would be helpful if the initial blood testing results and abnormal results (if any) were documented.

2. Was there any concern for disseminated infection in cases 1, 2 and 4? Did they have fungal blood cultures sent for testing or have any imaging studies done? I understand that patient #3 had CT brain and orbits imaging performed. It would be of interest to the readers to have these points documented.

3. Was itraconazole therapeutic drug monitoring performed for the patients in the case series? They all received systemic itraconazole therapy for sporotrichosis, so it would be interesting to note whether this was done and what drug levels were achieved. I understand that this is not usually considered for cutaneous disease, but may be pursued for mucosal disease. If it is feasible to include the itraconazole drug levels for the patients, I would also recommend including a part about this in the Discussion.

Minor Comments

1. It would be very informative if the study authors are able to include clinical pictures related to the ocular and lymphocutaneous lesions, or pictures of the slit lamp examination. It is understandable if they are not available due to patient confidentiality reasons.

2. Page 2 of 12 – I believe that the word parinaud should be capitalized per convention (given that it was first described by Dr. Henri Parinaud).

3. Page 2 of 12 – I believe that per convention, the abbreviation for matrix assisted laser desorption ionization-time of flight mass spectrometry should be MALDI-TOF (and not MALTI TOF as stated).

4. Page 3 of 12 – in case 3, it would be educative to include the last glycosylated hemoglobin (HbA1c) level of the patient and whether he had underlying uncontrolled DM.

5. Page 4 of 12 – please include the relevant reference for the line “Studies conducted in Brazil confirmed zoonotic transmission from domestic and wild animals via penetration of skin or mucosa and inhalation”.

6. Page 8 of 12 – in the line “It is important to identify the sporothrix to the species level”, it would be better to state it as the “It is important to identify the Sporothrix isolate to the species level”.

Author Response

Dear reviewer,

I'm appreciate for your advice.

I will modify and adjust accordingly.

I will add the information of HIV and DM and Blood fungal culture of four patients into the manuscript. I will add the information of  third patient HbA1c level  as well.

The TDM level of itraconazole was not done for all four patients in view of TDM isn’t available in my centre.  This point will added in Discussion section.

I will add the citation reference of Studies conducted in Brazil confirmed zoonotic transmission from domestic and wild animals via penetration of skin or mucosa and inhalation

I also will change the sentences in page 8 of 12 as your suggest.

Unfortunately I don’t have other pictures of ocular and lymphocutaneous lesion or pictures of slit lamp examination.

Thank you for your kind comments.

Regards,

Yeow 

Reviewer 2 Report

Dear Authors,

you have dealt with an interesting topic of study but the paper you have submitted, in my opinion, needs to be improved and modified.

I believe it is not enough to treat and describe only clinical cases, which are very interesting in themselves.

The review part is not very clear as it is described as a 'literature review' but then presented as a 'systematic review'. My advice is to remove it from the manuscript or use it for a new paper because inserted in this way, it is not consistent.

Finally, is it necessary to include photos of people affected by ocular sporotrichosis? Try to obscure them in some way or exclude them from the paper.

Here are some of my suggestions and corrections:

The type of paper presented by the authors must be inserted before the title. Please check and integrate.

Lines 1-2 - I suggest changing the Title to read: Mucosal sporotrichosis from zoonotic transmission: description of four case reports.

Line 3 - missing asterisk next to corresponding author name. Please add.

Line 43 - Please change to "Cases presentation", as four cases are described.

Line 75 - Table 1 should be moved after "Case 4" as it is representative of the resistance profiles of the strains isolated in the four cases.

Author Response

Dear reviewer,

I'm appreciate for your advice.

I will modify and adjust accordingly. The title of my manuscript is changed as your advice.

I will edit and obscure the photo as your suggested. 

Thank you for your kind comments.

Regards,

Yeow 

Round 2

Reviewer 2 Report

Dear Authors, all the changes reported by me have been made. Only the formatting improvements are missing but the editorial staff will take care of this.

Thank you for relying on my review.